# Statistical Methods for Auditing the Quality of Manual Content Reviews

**Xuan Yang, Andrew Smart, Daniel Theron**
Google LLC
`yangxuan@google.com, andrewsmart@google.com, dtheron@google.com`

## Abstract

Large technology firms face the problem of moderating content on their online platforms for compliance with laws and policies. To accomplish this at the scale of billions of pieces of content per day, a combination of human and machine review are necessary to label content. Subjective judgement and human bias are of concern to both human annotated content as well as to auditors who may be employed to evaluate the quality of such annotations in conformance with law and/or policy. To address this concern, this paper presents a novel application of statistical analysis methods to identify human error and these sources of audit risk.

## 1 Background and Problem

Content moderation has been defined as 'governance mechanisms that structure participation in a community to facilitate cooperation and prevent abuse,' (Gorwa et al., 2020). Online platforms may use a combination of algorithmic and manual review methods to handle the volume of content they host (Binns et al., 2017). Audits of such methods may be required to ensure that they conform to regulatory, policy or other requirements (MacCarthy, 2020). During these audits, auditors check the accuracy of the moderation decisions against agreed-upon policies. The problem of inter-rater reliability during these audits may result in an increased source of risk against consistency and neutrality requirements, as human review is naturally subjective and it may be difficult to maintain or measure consistency and equality among independent reviewers (Keller et al., 2020; Neal, 2022). Legal risks may also arise if reviews are not systematically equal in some contexts (Ángel Díaz & Hecht-Felella, 2021). This paper evaluates quantitative methods to measure and to minimize audit risks as a result of human reviews via statistical analysis.

## 2 Methodology: Prospective Analysis

Consider a scenario where there is a need to determine if certain products are being marketed towards appropriate demographics, as some may require additional scrutiny during internal review. During experimentation, we obtained a synthetic data set consisting of three reviewers and their answers regarding a nine question rubric on 1,528 products (see the data and code in appendix). Our measure of reviewer consistency is the agreement rate: the percentage of products for which there is complete reviewer consensus, assuming there are multiple reviewers reviewing the same content with multiple criteria questions.

By calculating the agreement rate, we can determine which questions reviewers have difficulty reaching a consensus on and may represent a source of bias. One of the approaches we can use when there are multiple rubric review questions is Fleiss Kappa (McHugh, 2012), which is a statistical measure for assessing the reliability of agreement between a fixed number of raters. This is accomplished by analyzing the distribution of the reviewers' results, and the correlation of each rubric review question to the human labeled result. This approach can be applied at both the question level and sampled group level. For example, based on the Table 1, the increased level of Fleiss Kappa represents an increased level of agreement between reviewers on each question, thus question 3 has a systematically higher disagreement rate. This may indicate some experimental design error for this question, and allows us to quantify the level of dissimilarity of subject matter expert opinions, rather than relying on conventional subjective audit reviews.

Table 1: Fleiss Kappa result for subset of sampled review group

| Rubric Question | Fleiss Kappa | Overall Fleiss Kappa |
|---|---|---|
| Question 1 | 0.649255 | 0.475072 |
| Question 2 | 0.545445 | |
| Question 3 | 0.093997 | |
| Question 4 | 0.399111 | |

We wish to determine if there is a systematic relationship between each criterion or rubric question and the final reviewer classification. This can be done through a chi-square test. Based on our risk tolerance and generally accepted industry standards (Broderick, 1974), we test at the 5% alpha level, which if met would signify that there is 95% confidence to conclude that there is a relationship between a question's answer and the reviewer classification.

We also wish to determine if there is a systematic difference between review teams and the ground truth result for different products. In scenarios where there are only two review teams to compare across, this can be done through hypothesis testing with a t-test. In scenarios where there are more than two comparison groups, an ANOVA test is more appropriate to avoid multiple pairwise comparisons. This analysis allows us to determine if the difference between sampling error rates are systematically different between groups.

## 3 METHODOLOGY: RETROSPECTIVE ANALYSIS

Consider another scenario where reviewers conduct reviews sequentially instead of simultaneously. This may result in one reviewer being able to see previous reviewers' review results. There is existing research to suggest human error exists in single blind vs double blind settings (A Tomkins, 2017) even in simultaneous review cases. We want to evaluate whether sequential review scenarios and their blinding effects would also affect review quality.

We can design an experiment among two groups of reviewers where one group can see the previous reviewers' result and the other one cannot by running an A/B test. However, many audit scenarios might not allow for direct interdiction into the process being audited, or may be resource constrained. In such a scenario, a method called Difference-in-Difference may be employed, as it uses historical data to approximate the effects of review changes (Anders Fredriksson, 2019).

Consider two groups: pre and post review change. One of the groups has been subject to the review change, while the other has not. One can assume that these two groups would have behaved roughly identically were it not for the review change, and that any systematic differences can be attributed to the review change itself. By observing the difference in performance, we can quantify how big of a difference the review change had in these groups' behavior. Refer to Figure 1 in the appendix, where the solid lines represent actual, historical trends, and the dotted line refers to a hypothetical performance (the counterfactual) from the treatment group where no changes to review structure had occurred. The difference between the actual vs hypothetical performance in the treatment group represents the treatment effect.

## 4 CONCLUSION

The statistical techniques presented here help auditors identify high risk groups, question design choices, and review methods that are more likely to result in human review error. For example, these techniques allow auditors to identify which questions are most likely to have a high disagreement rate (Question 3), which reviewers are most likely subject to bias (Reviewer 1), the relationship between rubric questions and the final classification, and the agreement rate for the overall group (0.47). These insights are unlikely to be gleaned from traditional, more subjective audit methodologies, which would have considered all questions equally well formulated. This paves the way for remediation suggestions more precisely targeted to reduce human error and sources of experimental risk (Gino & Coffman, 2021). Through these methods, we are able to use statistical analysis and experimental design to quantitatively and pinpoint sources of human error in audit procedures.

URM STATEMENT

The authors acknowledge that at least one key author of this work meets the URM criteria of ICLR 2023 Tiny Papers Track.

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

## A   APPENDIX

1) The code and data for the prospective analysis discussed in this paper: https://github.com/xuanyang0607/openreviewpaper

2) Inter-rater reliability: is the level of consistency among individual raters. In this paper, we measure the inter-rater reliability through Fleiss' kappa.

3) Sample group represents the sampled products that are being reviewed during experimentation. Rubric questions represent the review criteria that the product is being reviewed upon.

4) Fleiss Kappa: Statistical measure for assessing the reliability of agreement between a fixed number of raters. Fleiss Kappa is defined as:

$$\kappa = \frac{\bar{p} - \bar{p}_e}{1 - \bar{p}_e}$$

Where $\bar{p}$ is the average reviewers' agreement and $\bar{p_e}$ is the sum of squares of the proportion of assignment to the index category.

5) Chi Square Equation: It is a statistical hypothesis testing that helps to identify if two categorical variables have significant relationship or not. Hypotheses:

$H_o$: Null hypothesis. Eg: p1 = p0
$H_a$: Alternative hypothesis. Eg: p1 ¡¿ p0

Null hypothesis represents the statement of a test that intend to show no effect or no relationships between the tested variables.

Alternative hypothesis is the statement the opposite of the null hypothesis which represents the statement that intends to show effect or relationship between the tested variables.

$$\tilde{\chi}^2 = \frac{1}{d} \sum_{k=1}^{n} \frac{(O_k - E_k)^2}{E_k}$$

Where:

d represents the degrees of freedom

O represents the observed value

E represents the expected value

Comparing the value with the chi-square distribution to assess whether to reject or fail to reject the null hypothesis statement.

6) T-test: A statistical test to determine if there is a significant differences between two groups.

Hypotheses:

$H_o$: Null hypothesis.
$H_a$: Alternative hypothesis.

$H_o : \mu \geq \mu_o$ vs $H_a : \mu < \mu_o$ (one-tail test, lower-tail)

$H_o : \mu \leq \mu_o$ vs $H_a : \mu > \mu_o$ (one-tail test, upper-tail)

$H_o : \mu = \mu_o$ vs $H_a : \mu \neq \mu_o$ (two-tail test)

Test Statistic:
Case 1: $\sigma^2$ is known
$z = \frac{\bar{x} - \mu_o}{\frac{\sigma}{\sqrt{n}}} \sim N(0, 1)$
Case 2: $\sigma^2$ is unknown
$t = \frac{\bar{x} - \mu_o}{\frac{s}{\sqrt{n}}} \sim t_{n-1}$

7) ANOVA testing: A statistical test to determine if there are significant differences between two or more groups.

Hypotheses:

$H_o : \mu_1 = \mu_2 = ... = \mu_k$
$H_a : \mu_i \neq \mu_j$ for $i \neq j$

Test Statistic:

$F_{obs} = \frac{\text{MSTr}}{\text{MSE}} \sim F_{v_1, v_2}$      $v1 = df(\text{SSTr}) = k - 1$
$v2 = df(\text{SSE}) = n - k$

Reject $H_o$ if $F_{obs} \geq F_{\alpha, v_1, v_2}$

$$\text{SSTr} = \sum_{i=1}^{k} \sum_{j=1}^{n_i} (\overline{y}_{i\cdot} - \overline{y}_{\cdot\cdot})^2 = \sum_{i=1}^{k} \frac{1}{n_i} y_{i\cdot}^2 - \frac{1}{n} y_{\cdot\cdot}^2$$

$$\text{SSE} = \sum_{i=1}^{k} \sum_{j=1}^{n_i} (y_{ij} - \overline{y}_{i\cdot})^2 = \sum_{i=1}^{k} \sum_{j=1}^{n_i} y_{ij}^2 - \sum_{i=1}^{k} \frac{y_{i\cdot}^2}{n_i} = \sum_{i=1}^{k} (n_i - 1) s_i^2$$

8) Further Analysis for Prospective Analysis:

It may also useful to extrapolate the sampling error rate to the population by calculating the confidence interval of the error rate using the binomial distribution. This helps auditors quantify the impact of human reviewer bias, and identify high risk groups of products for further investigation.

Lastly, in order to help identify to what extent various factors might affect reviewer bias, we can use both a regression model, as well as a classification model to determine which metrics (ex.: product metadata or rubric criterion) have the most influence on the classification results by leveraging the beta coefficients. In general, a classification models should give similar metrics as regression models regarding the cause of bias.

9) Figure 1 Difference in Difference approach

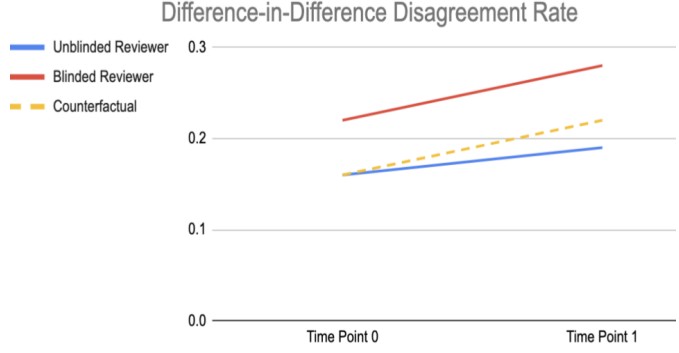

Figure 1: Difference in Difference approach

10) Figure 2 Methodology Flowchart

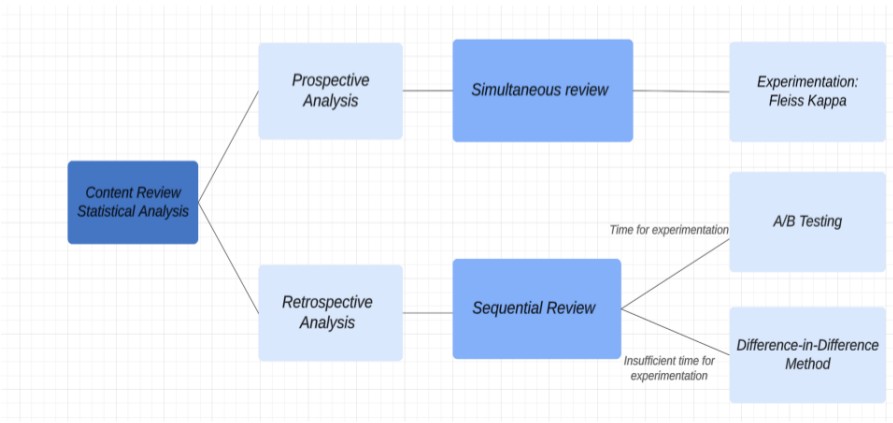

Figure 2: Methodology Flowchart

