# OpenReview forum: "Statistical Methods for Auditing the Quality of Manual Content Reviews"
_ICLR.cc/2023/TinyPapers — Submitted to Tiny Papers @ ICLR 2023_

### Official Review · Reviewer_MLWb · 2023-03-29

**Confidence:** 3

**Summary Of Contributions:**

This paper presents several statistical analysis methods to help researchers and auditors identify problematic human reviews/annotators. These methods include prospective analysis (chi-square test, t-test, and ANOVA test) as well as retrospective analysis (experiment design, difference-in-difference)

**Rating:**

Clear, Correct, and Reproducible (CCR): a submission which meets the reviewing criteria

**Strengths And Weaknesses:**

## Strengths and weaknesses

#### Clarity

1. Overall the paper is clear and easy to read.
2. What is "inter-rater reliability"?

#### Correctness

1. The proposed statistical methods look sound and useful, but there is no experiment to evaluate the proposed methods.

#### Reproducibility

1. This paper proposes several statistical methods to evaluate human reviews. It discusses these methods in detail in sections 2 and 3. The appendix also includes the formula for the proposed methods.

#### Follows basic requirements

1. This paper follows the basic formatting requirements.


**Suggested Changes:**

## Suggested changes

1. It would be great if the paper's main text can include a formula to compute Fleiss Kappa score, as it might be unfamiliar to many readers.
2. Table 1 needs to explain what "sample groups" and "rubric questions" are.

---

### Official Review · Reviewer_uJcE · 2023-03-30

**Confidence:** 3

**Summary Of Contributions:**

The paper proposes to use statistical tools to identify disagreements introduced during auding with human reviews.

**Rating:**

Needs Clarification (NC): a submission which does not meet the reviewing criteria and needs clarification for its described problem or solution

**Strengths And Weaknesses:**

Strengths:
1. The paper proposes multiple scenarios and factors that could potentially impact the reviewer's decision choice and might introduce a source of bias in evaluation, for ex: question criteria, sampling bias.

Weaknesses:
1. While the paper presents the usefulness of the tools to determine the source of agreement, there are no results that grounds their hypotheses.

**Suggested Changes:**

It is unclear what has been studied and what is to be studied/proposed. It is also unclear what the source of the dataset for the analysis is.

---

### Author Response · Authors · 2023-05-30
**Paper archival**

We wish to opt-in for archival.

Thank you so much for all your feedback, we have incorporate all the feedbacks into our revision, really appreciate all the feedback.

---

### Meta-Review · Area_Chair_jj6K · 2023-04-08

**Recommendation:** Invite to archive
**Confidence:** 3

**Metareview:**

1. Clarity: author has communicated clearly with relevant literature
2. Correctness: justification missed in the conclusion
3. Reproducibility: code & data not provided
4. Pros:
     * novel application to reduce human bias & judgement
5. Cons:
    * spelling mistakes
    * gist or explanation of few unknown terms:  inter-rater reliability
    * fails to provide explanation of how well their novel application outperforms in the conclusion

**Summary:**

minimizing audit risks by statistical analysis, however fails to provide comparison of how well it performs

**Comments And Feedback To The Authors:**

Authors must make few revisions to the paper to be CCR
1. kindly do a spell check in the document, analysis is misspelled in section 4
2. Include the code and data in the appendix
3. Considering tiny papers intended audience, its best to provide in the appendix the different tests(t-test,ANOVA test, A/B test, chi-square test) & relevant explanation
4. Provide experimental results that support your assumptions of your approach
5. Performance comparison should be in place on your proposed novel application.
    * How well your proposed novel approach performs compared to normal methodology should be mentioned.
6. In appendix, a flowchart of your methodologies in section 2, 3, will help understand better.

**Reason For Not Giving A Higher Recommendation:**

1. clearer conclusion is required
2. paper lacks to provide sufficient information on how well their novel application outperforms

**Reason For Not Giving A Lower Recommendation:**

1. The paper can be made stronger with revisions as suggested

---

### Decision · Program_Chairs · 2023-04-10

Revision accepted; invite to archive